# Genome-Wide Identification of the TIFY Family in *Cannabis sativa* L. and Its Potential Functional Analysis in Response to Alkaline Stress and in Cannabinoid Metabolism

**DOI:** 10.3390/ijms26178171

**Published:** 2025-08-22

**Authors:** Yuanye Zhang, Ming Zhang, Yuyan Fang, Nan Zheng, Bowei Yan, Yue Sui, Liguo Zhang

**Affiliations:** Institute of Industrial Crops, Heilongjiang Academy of Agricultural Sciences, Harbin 150086, China; zhangyyunity@outlook.com (Y.Z.); 18845874548@163.com (M.Z.); zn7225m@163.com (N.Z.); ybw1625wby@163.com (B.Y.);

**Keywords:** *Cannabis sativa* L., TIFY gene, transcription factors, alkaline stress, cannabinoid metabolism

## Abstract

TIFY transcription factors play crucial regulatory roles in secondary metabolism and stress response. However, the expression patterns of the *Cannabis sativa* L. TIFY gene family under alkali stress, their involvement in cannabinoid metabolism, and their underlying genetic evolutionary mechanisms remain largely unexplored. In this study, we used bioinformatics approaches to conduct genome-wide identification and functional characterization of the *C*. *sativa* TIFY gene family. Fourteen TIFY genes were identified and mapped onto seven chromosomes. These genes were classified into four subfamilies: TIFY, JAZ, ZML, and PPD, with the JAZ subfamily further subdivided into five distinct branches. Collinearity analysis suggested that gene duplication events contributed to the expansion of the TIFY gene family in *C*. *sativa*. Weighted gene coexpression network analysis (WGCNA) revealed that *CsJAZ2*, *CsJAZ3*, and *CsJAZ6* participated in the cannabinoid regulatory network. Cis-element analysis indicated that the promoter regions of TIFY genes were enriched in hormone- and stress-responsive elements. Furthermore, transcriptome and RT-qPCR analyses were conducted to examine gene expression patterns under alkaline stress (the RNA employed in RT-qPCR was extracted from the apical leaves of samples subjected to short-duration alkaline stress treatment). The results showed that *CsJAZ5* and *CsJAZ6* were downregulated, whereas *CsPPD1*, *CsTIFY1*, and *CsZML1* were upregulated in response to alkali stress. In summary, *CsJAZ5*, *CsPPD1*, and *CsTIFY1* may serve as candidate genes for the development of alkali-tolerant cultivars, while *CsJAZ2* and *CsJAZ3* may be valuable targets for enhancing cannabinoid production. This study provides important molecular insights and a theoretical basis for future research on the evolutionary dynamics and functional roles of TIFY transcription factors, particularly in stress adaptation and cannabinoid metabolism.

## 1. Introduction

The TIFY family comprises plant-specific transcription factors characterized by conserved TIFY motifs and is involved in a range of developmental, metabolic, hormone stimulation, and stress response processes. The gene *AT4G24470* was first identified in *Arabidopsis thaliana* and was initially annotated to contain a ZIM domain. Subsequent studies revised this to a TIFY domain, reflecting an updated understanding of its conserved motif structure [1,2]. TIFY transcription factors are categorized into four subfamilies based on their evolutionary relationships and conserved domains: TIFY, JAZ, PPD, and ZML. The TIFY subfamily contains only the TIFY domain, the JAZ subfamily contains a C-terminal jasmonate–ZIM domain, the PPD (PEAPOD) subfamily includes an N-terminal PPD domain, and the ZML subfamily contains both CCT and GATA domains. Notably, the PPD domain remains poorly characterized and lacks a clearly defined sequence signature [3]. Following *A. thaliana*, TIFY genes have been identified in several plant species, including cotton (*Gossypium arboretum*, *Gossypium raimondii*, *Gossypium hirsutum*) [4], *Vitis vinifera* [5], *Eriobotrya japonica* [6], *Manihot esculenta Crantz* [7], *Artemisia argyi* [8], and *Phalaenopsis aphrodite* [9], among others. In wheat, 63 *TIFY* members with a large genome size were identified, while 16 *TIFY* genes were discovered in pomegranate [10]. The JAZ subfamily consistently emerged as the most abundant group. For example, 24 PtTIFY genes were identified in *Populus trichocarpa*, half of which belonged to the JAZ subfamily [11]. In *Brassica napus* L., 142 TIFY genes were identified [12], including 105 JAZ members, confirming the numerical dominance of this subfamily.

TIFY family members exhibit diverse tissue-specific expression patterns. In luffa, *LcTIFY7* and *LcTIFY13* were predominantly expressed in the reproductive organs, whereas *LcTIFY11* showed higher expression in female than in male flower buds [13]. In tomato, *NtTIFY1*, *NtTIFY4*, *NtTIFY18*, and *NtTIFY30* were preferentially expressed in roots, whereas *NtTIFY7* was consistently expressed in plants inoculated with *P. aquatica* [14]. *AtZML1* and *AtZML2* are implicated in the hypocotyl growth regulation, with the *AtZML1* overexpression producing elongated petioles and hypocotyls [1]. *AtTIFY4a* (*PPD1*) and *AtTIFY4b* (*PPD2*) are involved in synchronized leaf growth [15]. *PPD2* forms a repressive complex with *LHP1* to regulate lateral organ growth [16]. *PPD2* also activates *JAZ7/8* to positively regulate plant defense, promote root development, and enhance salt stress tolerance [17]. Furthermore, PPD modulates light signaling by inhibiting SPA1, thereby affecting photomorphogenesis [18]. Deletion of *VmPPD* has been shown to enhance grain yield via increased seed size [19]. These findings suggest that TIFY transcription factors participate in a variety of physiological pathways.

JAZ proteins represent the most abundant subgroup within the TIFY family and function as key repressors of the JA signaling pathway. In *A. thaliana*, 13 JAZ genes have been identified and classified into five phylogenetic groups: Group I (*JAZ1/2/5/6*), Group II (*JAZ10*), Group III (*JAZ11/12*), Group IV (*JAZ3/4/9*), and Group V (*JAZ7/8/13*) [20]. *AtJAZ1* represses the transcription of JA-responsive genes, and elevated JA levels promote its interaction with SCFCOl1, leading to proteasomal degradation via the 26S proteasome pathway [21]. Members of Group IV (*AtJAZ3/4/9*) have been shown to suppress insect defense responses and alleviate JA-induced delays in flowering. Group III proteins (*AtJAZ10/11/12*) mitigate root growth inhibition by attenuating JA signaling [22]. *AtJAZ7* plays a role in immune defense against *Pseudomonas syringae* by modulating ROS homeostasis, energy balance, and glucosinolate biosynthesis. In the context of *F*. *albicans* infection, *JAZ7* inhibits JA overaccumulation through homo- and heterodimer formation. The overexpression of *JAZ7* has also been reported to enhance drought tolerance [23,24]. In barley, *HvJAZ2* acts as a negative regulator of drought tolerance by inhibiting root development, whereas *jaz2* mutants exhibit the upregulation of root development-related genes, including *SHR1*, *PLT1*, *PLT2*, and *PLT6*. In maize, *ZmJAZ13* was cloned and found to be highly expressed in immature embryos. Its heterologous expression in *A. thaliana* led to increased chlorophyll content, elevated peroxidase activity, and improved resistance to drought and salt stress [25]. In *Salvia miltiorrhiza*, the JA-responsive repressors *SmJAZ3*, *SmJAZ4*, and *SmJAZ8* interact with *SmMYC2* and *SmMYB36* to modulate tanshinone accumulation in a JA-dependent manner. In tobacco cultivar TN90, 32 TIFY genes were identified from whole-genome data, among which five TIFY/JAZ genes were differentially expressed in COI1-deficient plants and were found to influence the biosynthesis of secondary metabolites [26]. In longan, *DlTIFY7* negatively regulates anthocyanin biosynthesis [27]. Collectively, these findings demonstrate that TIFY transcription factors are involved in a wide range of physiological and molecular processes, including plant growth and development, flowering regulation, secondary metabolite biosynthesis, insect defense, root architecture formation, and responses to salinity and drought stress.

Soil salinization is a global environmental issue that impedes economic development. Globally, the total area of saline–alkali land has reached 1.381 billion hectares [28]. The Songnen Plain in China, with 3.43 million hectares of saline–alkali land, is one of the three major concentrated regions of saline–alkali land in the world [29]. Heilongjiang Province, located on the Songnen Plain, is one of the two provinces in China where hemp cultivation is permitted [30,31]. The area of saline–alkali land in the province accounts for 7% of its total cultivated land [32]. More importantly, as an economic crop, hemp cannot be grown on prime farmland in China, and thus is required to be planted on non-prime farmland. Non-prime farmland includes general farmland, forest land, sloping land, low-lying land, and the aforementioned saline–alkali land. However, general farmland accounts for a very small area. Therefore, hemp can typically only be grown on such lands, which are of lower quality than prime farmland [33]. Abiotic stresses such as saline–alkali stress and drought stress, which may occur in non-prime farmland, are key factors limiting the yield increase of industrial or medicinal hemp. Therefore, it is imperative to provide guidance for the breeding of stress-resistant *C. sativa* varieties by deeply analyzing the molecular mechanism underlying *C. sativa* stress resistance. Alkaline stress and salt stress disrupt the ionic homeostasis and water potential balance of plant cells, causing osmotic imbalance and ionic damage that ultimately lead to plant poisoning or even death [34]. Furthermore, the high pH stress and CO_3_^2−^/HCO_3_^−^ stress inherent in alkaline stress will further accelerate these processes [35]. Alkaline stress causes greater harm to the seedlings of *C. sativa*. Plants respond to alkaline stress through multiple signaling pathways. Signals are transmitted via a series of transcription factors (e.g., TIFY transcription factors) [10,12], which induce the production of certain protective proteins and metabolites. Among these stress-responsive metabolites in *C. sativa*, cannabinoids are a key type. Some studies have confirmed that TIFY transcription factors play a significant role in the response to alkaline stress. Zhu et al. found that *GsTIFY10* in *Glycine max* can be induced by bicarbonate, salt stress, and JA, and that this gene also inhibits the transmission of JA signals. In *A. thaliana*, overexpression of *GsTIFY10* enhances tolerance to alkaline stress during stages including seed germination, early seedling growth, and the adult plant stage [36]. Furthermore, they demonstrated that overexpression of *CsJAZ2* in *A. thaliana* enhances tolerance to saline–alkali stress [37]. Research on the biological role of *CsTIFY* in the context of alkaline stress is scarce and requires further exploration.

*C. sativa* is an annual herbaceous species of the genus *Cannabis* within the Moraceae family. It has a long history of global cultivation and is widely utilized for textile, industrial, and medicinal purposes [38,39]. Cannabidiol (CBD) is one of the major cannabinoid constituents that exhibits antagonistic effects against tetrahydrocannabinol (THC) [40,41]. It has been extensively applied in modern medicine and demonstrated to possess anti-inflammatory, analgesic, antiepileptic, antiasthmatic, anticancer, sedative, and dermatological properties [42,43,44,45,46]. As a high-value medicinal compound, the enhancement of CBD and other cannabinoid yields remains a key objective [47,48]. The environment has a significant impact on cannabinoid content. As a crucial plant nutrient, nitrogen promotes plant growth and development when supplied in excess, but reduces cannabinoid production [49]. An appropriate nitrogen supply (10–30% NH_4_^+^) can support plant growth without impairing secondary metabolite production [50]. An increase in phosphorus concentration increases biomass but reduces cannabinoid concentration [51]. Salinity also affects cannabinoid accumulation: when NaCl concentration is ≤13 mM, cannabinoid content in ECs peaks at 1.8 and 3 mS·cm^−1^ [52]. Furthermore, during the final stage of medicinal cannabis growth, flushing the growing medium can enhance cannabinoid production [53]. Canopy-level photosynthetic photon flux density (PPFD) is an important indicator for quantifying light intensity. At a PPFD of 1800 μmol·m^−2^·s^−1^, apical inflorescence density and harvest index peak are measured [54]. Therefore, in a closed system, adding an appropriate amount of nutrients and regulating light levels properly can enhance cannabis yield, while even mild stress can promote cannabinoid production.

Although the metabolic pathways of cannabinoids have been extensively studied, their regulatory mechanisms remain poorly understood [55,56]. Alkaline stress represents one of the primary abiotic stresses that *C. sativa* plants encounter when cultivated on non-prime farmland [30]. Research on the mechanisms underlying alkali stress tolerance in *C*. *sativa* is also extremely limited, and the role of the *C. sativa* TIFY family in alkali stress response has not been reported [31]. TIFY transcription factors may participate in multiple physiological and molecular processes and are also implicated in the regulation of high-value metabolite biosynthesis, including cannabinoids. However, limited information is currently available regarding the TIFY gene family in *C. sativa*. Therefore, we performed a genome-wide analysis, predicted the functions of the TIFY family using public transcriptome data, and treated cannabis seedlings with alkali to investigate the expression pattern of TIFY under alkaline stress. The investigation of this gene family holds paramount significance for elucidating the mechanism underlying the alkaline stress response and the regulatory network governing cannabinoid accumulation in *C. sativa.* Such an understanding can provide a theoretical foundation for breeding stress-tolerant and high-cannabinoid-yielding cultivars.

## 2. Results

### 2.1. Physicochemical Properties and Evolutionary Relationship Analysis of TIFY Transcription Factors

A total of 14 TIFY genes were identified in *C. sativa* through simultaneous alignment using BLAST and HMM models. These genes were named sequentially, based on their order of identification. A phylogenetic tree was constructed using 14 CsTIFY proteins from *C. sativa* and 19 AtTIFY proteins from *A. thaliana* (Figure 1). In *A. thaliana*, TIFY transcription factors were classified into four subfamilies: JAZ, PPD, ZML, and TIFY. Correspondingly, all four subfamilies were also identified in *C. sativa*, comprising nine JAZ members, four ZML members, and one member each from the TIFY and PPD subfamilies. Among the identified CsTIFY proteins, CsJAZ8 possessed the shortest coding sequence (196 amino acids) and the lowest molecular weight (17,524.89 Da), whereas CsTIFY1 exhibited the longest coding sequence (442 amino acids) and the highest molecular weight (46,479.54 Da). Five members, including four ZML proteins and CsJAZ6, were predicted to be acidic, whereas the remaining proteins were classified as basic. Based on the five established JAZ groups in *A. thaliana*, the JAZ subfamily members in *C. sativa* were classified as follows: Group I (*CsJAZ2* and *CsJAZ7*), Group II (*CsJAZ4*), Group III (*CsJAZ6*), Group IV (*CsJAZ1* and *CsJAZ5*), and Group V (*CsJAZ8*).

### 2.2. Gene Structure and Secondary Protein Structure of TIFY Transcription Factors

The motifs, conserved domains, and predicted secondary structures of TIFY proteins were integrated into a single figure alongside the phylogenetic tree. Gene structures of the TIFY family were mapped in conjunction with phylogenetic relationships. Motif composition, domain architecture, secondary protein structures, and exon–intron organization were displayed on the right side of the tree (Figure 2). All CsTIFY members contained motifs 1, 2, and 5, wherein motifs 2 and 5 overlapped with the conserved TIFY domain, while motif 1 corresponded to the JAS domain. The ZML subfamily exhibited distinct motifs 3 and 4, with motif 4 substituting the JAS domain by forming part of the CCT domain and motif 3 overlapping with the GATA domain. Additional motifs 7, 8, and 9 were uniquely present in CsJAZ2 and CsJAZ3, while CsJAZ1, CsJAZ5, and CsPPD1 featured motif 10. The TIFY domain represented the core conserved domain of the TIFY family and was verified in all 14 CsTIFY proteins. Within the conserved TIFY motif (QLTIFYGG), leucine and glycine residues following tyrosine were occasionally substituted by other amino acids in different subfamilies. In the ZML subfamily, substitutions were observed at positions corresponding to glutamine, isoleucine, phenylalanine, and glycine (Figure 2B). Three predominant secondary structures were predicted across the CsTIFY proteins: an alpha helix (3.79–21.78%), an extended strand (3.15–8.04%), and a random coil (73.01–93.06%) (Table 1). The maximum exon count was observed in CsZML3 and CsZML4, both containing nine exons, whereas the minimum exon number was found in CsJAZ8, which harbored only three. Notably, CsJAZ2 and CsJAZ3 exhibited highly similar sequences and nearly identical gene and protein structures(Appendix A). However, slight differences were detected in the distribution of their secondary structure elements. Multiple alpha helices and extended strands were typically observed within the TIFY domain, whereas alpha helices were predominantly present in the JAS domain. Distinct motifs were associated with specific secondary structural elements (Figure 2A).

### 2.3. Chromosomal Localization of TIFY Transcription Factors

Fourteen TIFY genes were unevenly distributed across seven *C. sativa* chromosomes. Chromosomes were visualized using a gradient color scale indicating gene density, where blue represents high-density regions and yellow denotes low-density regions (Figure 2C). Chr1 harbored the highest number of TIFY genes, with four members (*CsTIFY2*, *CsTIFY3*, *CsTIFY6*, and *CsTIFY7*). Although *CsJAZ2* and *CsJAZ3* exhibited high sequence similarity, their physical locations on Chr1 were widely separated. Notably, *CsTIFY7* was positioned within a gene-sparse region on Chr1. Chr2 contained *CsZML2* and *CsZML4*, which were closely located. Chr3 carried only one gene, *CsJAZ1*, also situated in a low gene density region. Two genes, *CsJAZ4* and *CsJAZ5*, were identified in the upper arm of Chr4. No TIFY genes were detected on Chr5, Chr6, and Chr8. Chr7 contained *CsJAZ8*, whereas *CsPPD1* was located near the centromere of Chr9. Chr10 harbored *CsTIFY1* along with two members of the ZML subfamily, *CsZML1* and *CsZML3*, which were positioned in close proximity.

### 2.4. Cis-Acting Elements of TIFY in C. sativa

We extracted the 2000 bp sequence preceding *C. sativa*. The TIFY transcription factor promoter was extracted for cis-acting element prediction. After removing ambiguous or erroneous annotations, 243 cis-acting elements were identified (Figure 3A). Based on their functional classification, these elements were grouped into four categories: light-responsive, hormone-responsive, stress-responsive, and physiological regulatory elements. Among these, 86 cis-acting elements were hormone-responsive, 69 were light-responsive, 65 were stress-responsive, and 23 were related to physiological regulation. *CsJAZ1* contained the highest number of cis-acting elements (35), including the greatest number of light-responsive (10) and hormone-responsive (18) elements. Notably, 10 of the hormone-responsive elements were ABA-responsive. *CsZML4* possessed the fewest cis-elements (9), comprising four light-responsive elements, two ABA-responsive elements, two anaerobic-responsive elements, and one endosperm-specific element. Only three endosperm expression elements were identified in total, each located in *CsZML4*, *CsZML3*, and *CsTIFY1*, respectively. All TIFY genes contained hormone-responsive elements, whereas the types of hormone-responsive elements varied among genes, and no TIFY gene possessed the full complement of hormone-responsive elements.

Different *CsTIFY* genes exhibit varying types and numbers of adversity-responsive cis-elements. Anaerobic response elements are commonly present across *CsTIFYs*, with *CsJAZ4* having the highest number (9). Other types of adversity-responsive elements occurred less frequently. Seven *CsTIFY* members (*CsPPD1*, *CsJAZ1/5/7*, and *CsTIFY1*) harbored low-temperature response elements. Drought-responsive elements were identified in six members: *CsPPD1*, *CsJAZ1/4/6*, and *CsZML1/3*. Defense response elements were detected in five members: *CsJAZ1/5/6/7* and *CsTIFY1*. Physiological response elements were rare and appeared in only a few genes, with their types varying among members. Meristem-associated elements were found in *CsPPD1*, *CsJAZ2/3/7*, and *CsZML2*. Zein metabolism-related elements were enriched in the JAZ subfamily including *CsJAZ2/3/4/6/7*. Similarly, circadian regulatory elements were restricted to JAZ members (*CsJAZ4/5/7*). Endosperm expression elements were present in *CsTIFY1* and *CsZML3/4*. Cell cycle-associated elements were exclusively identified in *CsJAZ1*, whereas phytochrome downregulation elements were specific to *CsZML2*. Notably, a hypoxia-responsive enhancer was observed only in *CsJAZ2/3* and absent in all other members. The diversity and abundance of cis-acting elements suggest that TIFY transcription factors may participate in a wide range of physiological and molecular processes in *C. sativa*.

### 2.5. Collinearity and Protein Tertiary Structure of TIFY

Collinearity analysis was conducted between the genomes of *A. thaliana* and *C. sativa* based on the sequences of the TIFY gene family. Ten collinear relationships were identified between the two species (Figure 3B). Among these, only three pairs exhibited one-to-one correspondence, whereas the remaining seven pairs displayed one-to-many or many-to-one relationships. Specifically, high collinearity and sequence similarity were observed in the following one-to-one pairs: *CsJAZ1* with *AtJAZ4*, *CsJAZ8* with *AtJAZ8*, and *CsZML1* with *AtTIFY1*. A high degree of homology was also observed between *CsJAZ2* and *CsJAZ3*, both of which exhibited strong similarity to *AtJAZ2*. In addition, *CsJAZ6* showed collinearity with *AtJAZ11* and *AtJAZ12*, whereas *CsJAZ7* was collinear with *AtTIFY5*, *AtTIFY6*, and *AtTIFY12*. Notably, *CsJAZ6*, *CsJAZ7*, and *AtTIFY12* shared high similarity and collinearity. Other *CsTIFY* genes displayed limited homology with their *A. thaliana* counterparts, whereas the alignment scores were insufficient to establish collinearity. Tertiary structure homology modeling of CsTIFY proteins was performed using the SWISS-MODEL database (Figure 4). The GMQE (Global Model Quality Estimation) values ranged from 0.45 to 0.64, supporting the credibility of the predicted structures. Among the 14 CsTIFY proteins, 9 exhibited 100% sequence identity with their templates, indicating high modeling reliability. For *CsJAZ2* and *CsJAZ3*, which shared the high sequence similarity, the identities from homology modeling differ by only one percent (52.14%, 51.06%), respectively, with identical GMQE values of 0.58. The homology modeling identities of JAZ2 and JAZ3 were relatively low, which means that functional predictions based on homologous proteins may be prone to errors. However, their Ramachandran plots showed favorable values (84.05% and 83.27%), indicating that their spatial conformations are reasonable. Minor structural differences were observed between their three-dimensional conformations. No significant structural similarities were found among the remaining CsTIFY proteins.

### 2.6. Expression Patterns of TIFY Transcription Factors in Alkali Stress and Inflorescence Tissue

To elucidate the potential functions of TIFY genes in *C. sativa*, two publicly available transcriptome datasets were analyzed prior to molecular biological validation. Under alkali stress, significant differential expressions were observed in eight *CsTIFY* genes. Members of the JAZ subfamily, including *CsJAZ4*, *CsJAZ5*, *CsJAZ6*, and *CsJAZ7*, were significantly downregulated. In contrast, members of other subfamilies such as *CsPPD1*, *CsTIFY1*, and *CsZML4* were significantly upregulated in response to stress (Figure 5A).

Additionally, inflorescence transcriptome data for *C. sativa* were retrieved from the GEO database. As the dataset lacked a control group, pairwise comparisons between cultivars were conducted. Transcriptome sequencing was performed on inflorescences of multiple *C*. *sativa* varieties, revealing significant differences in the expression of the TIFY family among different varieties. Specifically, the TIFY family expression levels in MT1-3 were generally lower than those in other varieties, while those in CC1-3 maintained an overall high trend. Additionally, the expression levels of *CsCBDAS* and *CsTHCAS* in BBK1-3 were lower than in other varieties, and the expression of the *CsOLS* gene in CT1-3 was significantly lower compared to other varieties. *CsJAZ2* and *CsJAZ3* emerged as transcription factors with marked variations in expression among cultivars (Figure 5B). The data used for WGCNA were derived from the inflorescence transcriptomes. Based on the transcriptomic profiles, all genes were classified into 17 coexpression modules, each assigned a distinct color. Correlation analysis between these modules and cannabinoid content was performed [57]. The results revealed a significant positive correlation between the blue module and the CBD/THC content. *CsJAZ2* and *CsJAZ6* were located in this module but were not identified as hub genes. The Magenta module, including *CsJAZ3*, exhibited a strong positive correlation with the CBG content. These findings suggest that, despite the high sequence similarity between *CsJAZ2* and *CsJAZ3*, these two genes may play distinct physiological roles (Figure 6).

### 2.7. Relative Expression Levels of TIFY Under Adverse Stress and MeJA Treatment

For the alkali stress condition (200 mmol/L Na_2_CO_3_), leaf samples were collected at 0, 12, 24, and 36 h, and the relative gene expression levels were quantified by RT-qPCR. Under 200 mmol/L Na_2_CO_3_ treatment, the expression levels of *CsJAZ1* and *CsJAZ8* were reduced, whereas *CsJAZ5*, *CsJAZ6*, and *CsZML2* were significantly downregulated (Figure 7A). In contrast, the expression of *CsJAZ2*, *CsJAZ3*, and *CsJAZ7* increased, as did the expression of *CsPPD1*, *CsTIFY1*, *CsZML3*, and *CsZML4*, with *CsZML3* displaying a particularly significant upregulation. Under alkaline stress, both genes exhibited transient downregulation at 12 h, followed by progressive upregulation at later time points (Figure 7C). The observed divergent expression patterns among the seven JAZ subfamily members under alkaline stress suggest potential functional redundancy within this subfamily. In addition, the expression levels of four genes involved in the cannabinoid metabolic pathway (*CsOLS*, *CsCBGAS*, *CsCBDAS*, and *CsTHCAS*) were analyzed under alkali stress. The relative expression levels of *CsOLS* and *CsCBGAS* increased significantly, whereas that of *CsCBDAS* showed an increasing trend that was not statistically significant. Notably, *CsTHCAS* expression was undetectable in all biological replicates (Figure 7B).

Furthermore, based on functional prediction and transcriptome profiling, *CsJAZ5*, *CsJAZ8*, *CsPPD1*, and *CsTIFY1* were selected for further expression analyses under three stress conditions (Appendix A). *CsJAZ8* was significantly upregulated under drought stress and exhibited a decreasing, although not statistically significant, trend under alkaline stress. *CsJAZ5* was significantly downregulated under all three stress conditions, and its expression under alkaline stress displayed a gradual decline over time. *CsPPD1* and *CsTIFY1* shared the similar expression dynamics, showing the marked upregulation under all stress treatments (Appendix A). To investigate the response of JAZ genes to JA signaling, the expression patterns of eight JAZ genes were examined under 50 mmol/L MeJA treatment. Except for *CsJAZ1*, which showed reduced expression, all other JAZ genes were upregulated. All changes were statistically significant, except for *CsJAZ6*, which showed no significant differences (Appendix A).

## 3. Discussion

### 3.1. Classification of the TIFY Family in C. sativa

TIFY is a plant-specific transcription factor that was officially named by Vanholme in 2007 and subsequently categorized into four subfamilies [2]. Following its initial characterization in *A. thaliana* [3], the molecular biological functions of TIFY family members have been extensively investigated in various plant species. For instance, 33 TIFY genes have been identified in tobacco, comprising 20 JAZ, 4 PPD, 7 ZML, and 2 TIFY members [14]. In cassava, 28 TIFY genes have been reported, including 16 JAZ, 3 PPD, 7 ZML, and 2 TIFY genes [7]. As a key repressor of the JA signaling pathway, the JAZ subfamily constitutes a substantial portion of the TIFY family and often accounts for nearly half of its members [8]. In this study, 14 TIFY genes were identified, of which eight belonged to the JAZ subfamily, reflecting a similar trend. Initially, *CsPPD1* was classified as a member of the JAZ subfamily based on the presence of a TIFY domain and a truncated JAS domain (Figure 1). However, subsequent phylogenetic analysis revealed that *CsPPD1* was more closely related to the PPD subfamily in *A. thaliana* and clustered within the same major branch. Therefore, *CsPPD1* was reclassified as a member of the PPD subfamily and renamed accordingly. In *A. thaliana*, the JAZ subfamily is divided into five branches, and in *C. sativa*, JAZ is also divided into five branches. The *AtTIFYs* in these branches correspond to previous studies [1,2], confirming the accuracy of the phylogenetic tree (Figure 1).

### 3.2. Genetic Structure and Classification of TIFY in C. sativa

The three subfamilies, JAZ, ZML, and TIFY, were clearly defined, and their domains were verified using the CDD search in NCBI. In contrast, the PPD subfamily lacks a well-defined motif and is commonly described as possessing a PPD domain at the N-terminus, which is not annotated in the major protein domain databases. Consequently, CDD search did not support the identification of the PPD domain. In a previous study, Bai et al. reported that models of the PPD domain could not be retrieved from Pfam or other databases during the analysis of TIFY families across 14 genomes [3]. Therefore, HMMER was employed to construct HMM profiles of the PPD domain for subsequent searches. Although additional PPD subfamily members were identified using this approach, the corresponding HMM model and conserved motifs have not been published. Thus, the structural and functional characteristics of the PPD domain remain unclear.

After SMART validation, the PPD domain of *CsPPD1* was not identified. However, two low-complexity regions were predicted at positions 94–106 aa and 280–296 aa. Motif analysis indicated that CsJAZ1, CsJAZ5, and CsPPD1 contained an N-terminal Motif10, corresponding to the base sequence “YFKGKGMQWPFSNKVSALPQLJSFKAPQD”. These results suggest that the sequences within the PPD subfamily have the potential for further investigation to elucidate the PPD domain. Comprehensive genome data related to *C. sativa* have just been released. We will further explore the characteristics of the PPD subfamily using higher-quality comprehensive genome data [58]. Interestingly, Motif2/5 was identified as a conserved element within the TIFY domain, whereas Motif1 was identified as a conserved element of the JAS domain (sequence: RKASLQRFLEKRKER). These motifs constituted the fundamental structural components of the TIFY family and were detected in nearly all TIFY transcription factors (with CsTIFY1 containing only Motif2/5). Motif2 itself harbored a highly conserved TIFY motif of 15 amino acids (QLTIFYGGQVYVFDD), whereas Motif5 immediately contiguous with Motif2 encompassed 13 amino acids (PPDKAQAILLLAG). Both motifs spanned almost the entire TIFY domain. Within the ZML subfamily, Motif1 did not participate in JAS domain formation, whereas it constituted the CCT domain in conjunction with Motif4, with the JAS and CCT domains sharing similar N-terminal sequences. Additionally, Motif3 overlapped with the GATA domain, effectively covering its core structure of the GATA domain and comprising a long motif of 46 amino acids (Figure 2A–C).

The conserved motif represents a critical structural feature required for transcription factor functionality, and the activity of JAZ proteins is primarily dependent on the presence of the JAS domain. The JAZ family in *C. sativa* exhibits a relatively conserved structure, suggesting its essential role in JA signaling. *CsJAZ2* and *CsJAZ3* constituted a unique gene pair, demonstrating nearly identical gene sequences, conserved motifs, and protein structures with only minor variations(Appendix A). These two genes are located at different positions on Chr1 and may have originated from transposable element activity. Moreover, *CsJAZ2* and *CsJAZ3* were found to exhibit distinct physiological and molecular functions.

### 3.3. Cis-Acting Elements of the TIFY Family

To facilitate visualization of the quantity and classification of cis-acting elements, a position map of cis-acting elements was integrated with a stacked bar chart (Figure 3A). Owing to the distinct chromosomal distributions of TIFY members, substantial variation was observed in both the type and number of cis-acting elements within their promoter regions. These elements are categorized into four functional classes: light-responsive, stress-responsive, hormone-responsive, and physiologically regulatory. Light-responsive elements were found to be ubiquitously distributed in the promoter regions of all *CsTIFY* genes, reflecting the essential role of photoreception in plant autotrophic growth [59]. Notably, clusters of densely localized light-responsive elements were detected in specific members. For instance, the six photoresponsive elements in *CsPPD1* were concentrated near the 3′ end, consistent with the established role of PPD in regulating leaf and organ development [15,16,18]. Dense light-responsive elements may contribute to photomorphogenic regulation. *CsJAZ1* and *CsJAZ5* possessed the highest number of light-responsive elements (more than 10), predominantly located in the central region of the promoter. Similarly, the promoter region of the *Artemisia argyi* JAZ family has been shown to be enriched with light-responsive elements and may be subject to light-mediated regulation [8]. The seven light-responsive elements in *CsZML3* are primarily situated near the 5′ end. Other *CsTIFY* members exhibited fewer photoresponsive elements. This observation is consistent with the findings of Zhao et al., who reported a relatively high abundance of ABA-responsive elements in the TIFY family of *Fagopyrum tataricum* [60]. In grape, TIFY transcription factors have been confirmed to respond to MeJA and ABA but not to SA or Eth [5]. *CsTIFYs* are predicted to contain anaerobic induction elements, whereas other stress-responsive elements are relatively scarce. Notably, *CsJAZ2* and *CsJAZ3* contain anaerobic response enhancers, which may function under hypoxic conditions, such as flooding. The diversity of cis-acting elements suggests that *CsTIFY* possess a broad range of potential molecular biological functions.

### 3.4. Homology of the TIFY Family in C. sativa and A. thaliana

*A. thaliana* AT4G24470 (*AtTIFY1*) was the first member of the TIFY family to be annotated, initially designated as ZIM. Subsequent studies led to the identification and functional investigation of additional TIFY members [1,2]. However, functional characterization of TIFY transcription factors in *C. sativa* remains limited, and their biological roles require further elucidation. In early phylogenetic analyses, *AtZML1/2* and *AtTIFY1* were clustered together, indicating a close evolutionary relationship (Figure 1). In this study, *AtTIFY1* clustered more closely with the ZML subfamily, whereas *CsTIFY1* exhibited close phylogenetic proximity to *AtTIFY8* (Figure 1). Moreover, *CsZML1* demonstrated both phylogenetic relatedness and collinearity with *AtTIFY1* (Figure 3B). Notably, *CsZML1* was significantly downregulated in response to Leptosphaeria and Mucor infection, suggesting its potential role in pathogen defense. Whether *CsZML1* performs functions analogous to those of *AtZML*, such as regulation of hypocotyl elongation and photomorphogenesis [15], remains to be determined. In addition to *CsZML1*, several members of the JAZ subfamily (*CsJAZ1/2/3/6/7/8*) exhibited collinearity with *A. thaliana* JAZ genes. Among them, *CsJAZ8* was collinear with *AtJAZ8* and positioned in close proximity to the phylogenetic tree alongside *AtJAZ7* and *AtJAZ13* (Figure 1). In the study by Yan et al. [61], *JAZ8* was shown to form a complex with *JAV1* and *WRKY51* to suppress JA biosynthesis and maintain low JA levels. Upon herbivory, phosphorylation-induced degradation of the JAV1–JAZ8–WRKY51 complex via the 26S proteasome promotes JA accumulation, enabling effective insect defense responses. *CsJAZ8* may perform a comparable role in insect defense. Notably, *JAZ13* is the only member of the *A. thaliana* JAZ family not classified within the TIFY family [62]. No homologous transcription factor with high similarity to *JAZ13* was identified in *C. sativa*. However, *CsJAZ8* was co-clustered with *AtJAZ13* in the JAZ V branch (Figure 1).

### 3.5. Function Prediction of TIFY Transcription Factors

Under alkali stress, *CsJAZ4*, *CsJAZ5*, *CsJAZ6*, and *CsJAZ7* were significantly downregulated (Figure 5A), which may alleviate the repression of the JA signaling pathway and thereby facilitate the activation of downstream biological processes involved in stress response. Concurrently, significant upregulation of *CsPPD1*, *CsTIFY1*, and *CsZML4* may indicate the involvement in alternative physiological regulatory pathways. In *A. thaliana*, PPD and ZML genes were implicated in leaf expansion, petiole elongation, and lateral organ development [1,16]. The TIFY family exhibits significant differences in expression among various *C*. *sativa* varieties. Specifically, the TIFY family in MT1-3 shows lower expression levels compared to other varieties, and the total cannabinoid content is also the lowest. In contrast, the TIFY family in CC1-3 maintains an overall high expression trend, with its cannabinoid content remaining at a high level (Appendix A). In BBK1-3, the expression levels of *CsCBDAS* and *CsTHCAS* are lower than those in other varieties, yet its total cannabinoid content is moderate. For CT1-3, the expression of the *CsOLS* gene is significantly lower than that in other varieties, and its total cannabinoid content is at the bottom (Appendix A). The above findings are consistent with those of other studies, which have demonstrated that the expression of cannabinoid metabolism-related genes does not exhibit a direct proportional relationship with cannabinoid content. Instead, the accumulation of precursor substances shows a positive correlation with the increase in cannabinoid content [63,64]. In the present study, we observed a strong correlation between the TIFY family and the cannabinoid metabolic pathway. To further validate this association, we performed a WGCNA analysis, which confirmed that the *JAZ2/3/6* genes may be involved in the regulatory network of the cannabinoid metabolic pathway (Figure 6). We will conduct further research to explore in depth their roles in the metabolism of cannabinoids.

To systematically explore the biological functions of TIFY, we extended our analysis to additional high-quality transcriptome datasets (Appendix A). We characterized TIFY expression patterns across diverse tissue origins and treatment conditions, allowing for a comprehensive assessment of its functional roles in different physiological and experimental settings. *C. sativa* is a dioecious species, and sex-specific gene expression differences have been observed [65]. Notably, *CsJAZ6* was significantly upregulated in female plants, whereas *CsZML3* was significantly downregulated in male plants, suggesting its potential role in floral organ development and sex differentiation (Appendix A). In the dataset evaluating the effects of Trichoderma hamatum on drought tolerance in *C. sativa* (Appendix A), only *CsZML1* exhibited significant downregulation in both CON vs. T and CON vs. TDRT comparisons. Compared with other adverse conditions, the TIFY family plays a relatively minor role in drought stress. Following Sclerotinia sclerotiorum infection, significant downregulation of *CsZML1*, *CsPPD1*, and *CsJAZ6* was observed between 72 and 168 h post-inoculation. Following Sclerotinia sclerotiorum infection, significant downregulation of *CsZML1*, *CsPPD1*, and *CsJAZ6* was observed between 72 and 168 h post-inoculation. In *A. thaliana*, PPD and ZML genes were implicated in leaf expansion, petiole elongation, and lateral organ development [1,16]. The downregulation of these transcription factors may reflect pathogen-driven modulation of host growth strategies, possibly enhancing disease tolerance through energy redistribution and attenuation of JA-mediated defense suppression [21,22] (Appendix A).

### 3.6. RT-qPCR Analysis of Cannabinoid Metabolism Genes

To validate the results of the WGCNA-based association analysis, we examined the relative expression levels of four genes involved in different steps of the cannabinoid metabolic pathway [66,67] (*CsOLS*, *CsCBGAS*, *CsCBDAS*, and *CsTHCAS*). Among these, *CsOLS* (olivetol synthase) is a synthetase of the precursor substance olivetol [63], responsible for catalyzing the conversion of hexanoyl–CoA and malonyl–CoA to olivetol. This enzyme plays a pivotal role in initiating this pathway [68,69]. *CsCBGAS* (*CsPT*) is a key rate-limiting enzyme that catalyzes the conversion of olivetolic acid (OAC) and geranyl diphosphate (GPP) to cannabigerolic acid (CBGA), a precursor for the biosynthesis of various cannabinoids [70]. Additionally, *CsCBDAS* and *CsTHCAS* encode synthases responsible for producing CBDA and THCA, respectively [71,72]. Under alkali stress, the relative expression levels of *CsOLS* and *CsCBGAS* were significantly upregulated, indicating that the cannabinoid metabolic pathway is activated in response to stress. These results are consistent with previous findings that stress promotes the synthesis of secondary metabolites.

Although *CsCBDAS* expression was detectable, the actual (pre-normalized) expression level was very low, and no statistically significant difference was observed after normalization. Additionally, *CsTHCAS* expression remained undetectable in repeated experiments (Figure 7B). To reduce experimental variability, all RT-qPCR samples, including those for cannabinoid pathway genes, were collected from the same batch of seedlings. However, cannabinoid biosynthesis primarily occurs in the glandular hairs of leaves, and the number and maturity of glandular hairs in seedlings are considerably lower than those in inflorescence tissues [73,74]. Therefore, the lack of significant *CsCBDAS* expression may be attributed to the developmental stage and density of the sampled glandular hairs. However, the upregulated expression of *CsCBDAS* remained evident, suggesting that its expression may be pronounced in female flowers.

For legal compliance, the hemp selected in this study was a low-THC variety with a THC content far below 0.3%, meeting the national standards in both China and the United States [75,76]. The absence of detectable *CsTHCAS* expression may be attributed to the gradual loss of *CsTHCAS* genes during long-term breeding of low-THC cultivars. In the WGCNA analysis, the expression of *CsJAZ2/6* was highly correlated with CBD and THC levels. Under alkaline stress, *CsJAZ2* expression exhibited an upward trend, though the change was not statistically significant, whereas *CsJAZ6* expression was significantly downregulated. These results suggest that the regulatory roles of *CsJAZ2* and *CsJAZ6* in the cannabinoid biosynthetic network are complex. Additionally, *CsJAZ3* was positively correlated with the CBG levels, which was consistent with the RT-qPCR results showing an upregulation trend. Future studies should focus on elucidating the functional relationships between these genes. Furthermore, cannabinoid metabolism varies across different developmental stages, and we will continue to investigate these variations. Collectively, our findings support the presence of both positive and negative correlations between TIFY transcription factors and cannabinoid metabolism, paralleling their known roles in secondary metabolism regulation in other plant species [26,27].

### 3.7. RT-qPCR Analysis of TIFY Transcription Factors

To validate the function and transcriptomic analyses of TIFY family genes, RT-qPCR experiments were performed, yielding relative expression level data. Neither *CsJAZ4* nor *CsZML1* was detected in the repeated stress response assays, suggesting the potential spatiotemporal specificity of their expression. Notably, *CsJAZ2* exhibited consistently high expression levels under both CK and alkaline stress conditions, significantly exceeding those of the other genes. However, *CsJAZ2* was not significantly different after normalization. This indicates a possible role in baseline physiological growth. *CsJAZ2* shared high homology with *AtJAZ1* known to form transcriptional complexes involved in organ development in *A. thaliana* [22,77]. Therefore, *CsJAZ2* may perform analogous molecular functions. A pronounced reduction in *CsJAZ6* expression was observed under alkaline stress, corroborating its significant downregulation in the transcriptome dataset. This downregulation may mitigate the suppression of JA signaling, thereby enabling a more robust stress response [60]. Similarly, the decreasing expression trend of *CsJAZ5* aligned well with transcriptomic observations. Conversely, *CsPPD1*, *CsTIFY1*, and *CsZML4* were significantly upregulated in both RT-qPCR and transcriptome analyses. In cassava, *MePPD*, *MeTIFY* and *MeZML* also showed an upregulation trend under stress [7], similar to the above results. These genes may contribute to growth modulation and energy reallocation during stress adaptation 23. Additionally, we observed that CsJAZ2/3/7, which are clustered within the JAZ I clade, exhibited upregulated expression under both alkali stress and MeJA treatment (Figure 7A, Appendix A). The secondary and tertiary structural similarities of CsJAZ2/3/7 (Figure 4), along with their co-localization on chromosome Chr1, suggest potential functional redundancy.

Additionally, to test whether *C*. *sativa* JAZ genes could respond to JA, we sprayed 50 mmol/L MeJA and verified the gene expression levels. Except for *CsJAZ6*, the expression levels of the other seven JAZ genes showed very significant differences. Among them, the expression of *CsJAZ1* showed a downward trend, while that of *CsJAZ2/3/4/5/7/8* showed an upward trend. These results confirm that *C*. *sativa* JAZ genes are able to respond to JA signals, and a similar trend in response to JA signals was also verified in *VvJAZ* and *BnaJAZ* [5,12]. During the alkali treatment, we simultaneously conducted drought and salt–alkali treatments and selected several key genes for investigation. These genes exhibited consistent expression trends in both transcriptome profiles and RT-qPCR analyses (Appendix A). Notably, *CsJAZ8* expression was undetectable in the salt–alkaline treatment group, suggesting that other regulatory factors might have inhibited its transcription. Under drought stress, *CsJAZ8* expression increased significantly, showing a highly significant difference, whereas it trended downward under alkaline stress with no significant change. These findings indicate that *CsJAZ8* may play a crucial role in mediating drought stress responses. Multiple studies have also confirmed that JAZ is involved in the drought response [78]. *CsJAZ5* was significantly downregulated under both salt–alkaline and drought stresses, particularly in the former, implying its major function in salt–alkaline tolerance. *CsPPD1* and *CsTIFY1* displayed similar expression patterns under stress, with varying degrees of upregulation. Specifically, *CsPPD1* showed the most significant response to salt–alkaline stress (highly significant difference), while *CsTIFY1* exhibited the greatest change under drought conditions. Additionally, we analyzed the dynamic expression profiles of *CsJAZ5*, *CsPPD1*, and *CsTIFY1* from 0 to 36 h under alkaline stress. *CsJAZ5* expression decreased gradually over time, whereas *CsPPD1* and *CsTIFY1* showed a pattern of initial decrease followed by increase, confirming that alkali stress tolerance involves complex regulatory mechanisms. A notable observation is that, in other studies [7,79], members of the TIFY family also exhibited an expression pattern characterized by an initial decrease followed by an increase. This phenomenon can be attributed to several factors: due to functional redundancy within gene families, different family members may regulate distinct physiological processes (e.g., altering water conductivity, modifying tissue structure, or protecting enzyme activity). As stress intensifies gradually, one protective mechanism may become ineffective, while another related family member is strongly induced. Furthermore, regulatory mechanisms such as transcription factor phosphorylation and negative feedback loops may also influence the results obtained from RT-qPCR [80]. This fluctuating expression pattern reflects a complex physiological and molecular process, which we plan to explore through physiological and biochemical experiments in our subsequent research. As inhibitors of JA signaling, JAZ typically exhibit low expression under adverse conditions to facilitate JA-mediated defense responses. In contrast, subfamilies such as PPD, TIFY, and ZML often upregulate their expression to modulate plant physiological adjustments during stress [81,82]. Thus, this study explores the expression dynamics of the TIFY family under adverse environmental conditions. Notably, overexpression of *CsJAZ2* in Glycine soja has been demonstrated to enhance plant tolerance to salt–alkaline conditions. CsTIFYs have a rich expression pattern in response to alkaline stress, and they have the potential to improve salt–alkaline tolerance through overexpression [36].

## 4. Materials and Methods

### 4.1. Genome-Wide Identification of TIFY Transcription Factors in C. sativa

Whole-genome sequences of *A. thaliana* (GCF_000001735.4) and *C. sativa*. (GCF_900626175.2) were downloaded from NCBI (https://www.ncbi.nlm.nih.gov/datasets/genome/, accessed on 15 August 2024). The TIFY gene family sequences in *A. thaliana*, including *JAZ13*, were retrieved from NCBI (https://www.ncbi.nlm.nih.gov/gene/?term=TIFY, accessed on 15 August 2024). The Hidden Markov Model (HMM) profiles of conserved domains within the TIFY family were obtained from InterPro (https://www.ebi.ac.uk/interpro/, accessed on 15 August 2024), including the TIFY domain (PF06200), jasmonate domain (PF09425), and the CCT domain (PF06230). Subsequently, both “BLAST” (E-value < 1 × 10^−5^; NumofHits = 500; NumofAlignments = 250) and “Simple HMM Search” in TBtools (v2.154) [83] were used for sequence comparison. The intersection of the results from both methods was retained for further analysis. Preliminary domain verification was conducted using the Batch CDD Search tool (https://www.ncbi.nlm.nih.gov/Structure/bwrpsb/bwrpsb.cgi, accessed on 15 August 2024), and sequences lacking the required conserved domains were excluded. Genes with qualified domain structures were retained, and the final protein sequences of the *C. sativa* TIFY gene family were obtained.

### 4.2. Physicochemical Property Prediction, Chromosome Localization, and Evolutionary Analysis of TIFY Transcription Factor

ExPASy (https://web.expasy.org/protparam, accessed on 16 August 2024) was used to predict the physicochemical properties of protein sequences. Upon inputting each sequence, the number of amino acids, molecular weight, isoelectric point (pI), instability index, and grand average of hydropathicity (GRAVY) were obtained. Chromosome localization was visualized using the GTF/GFF-based location visualization tool in TBtools. Multiple sequence alignment of the protein sequences was conducted using MUSCLE Wrapper in TBtools (maximum iteration = 16, maximum hours = 0). The aligned sequences were subsequently trimmed using trimAL Wrapper in TBtools to remove excessively long regions. The trimming pattern selected was ML_AUTOMATED1. The resulting processed sequences were then used to construct a phylogenetic tree using the FastTree GUI Wrapper in TBtools. Final visualization and beautification of the phylogenetic tree were performed using TVBOT (https://www.chiplot.online/tvbot.html, accessed on 16 August 2024) [84].

### 4.3. Gene Structure and Collinearity Analysis of TIFY Transcription Factors

MEME (https://meme-suite.org/meme/tools/meme, accessed on 20 August 2024) was used to predict the conserved motifs within the TIFY protein family, with 10 motifs. The conserved motif patterns of CsTIFY proteins were subsequently visualized using Jalview Simple Launcher in TBtools to illustrate their shared and divergent features. The secondary and tertiary structures of TIFY proteins were predicted using NPS@SOPMA (https://npsa.lyon.inserm.fr, accessed on 5 September 2024) and SWISS-MODEL (https://swissmodel.expasy.org, accessed on 5 September 2024), respectively. During secondary structure prediction, the proportions of alpha helices, extended strands, and random coils were recorded and compiled in the same dataset, along with physicochemical properties and gene information. For collinearity analysis, the genome and annotation files of *A. thaliana* and *C. sativa* were inputted into the One Step MCScanX module in TBtools to generate whole-genome collinear blocks. The resulting collinearity data, along with the corresponding gene names, were then visualized using the Dual Synteny Plot function in TBtools.

### 4.4. Cis-Acting Element Prediction of TIFY Transcription Factors

The promoter sequences (2000 bp upstream) of CsTIFY genes were extracted using the GTF/GFF3 Extract function and submitted to PlantCARE (https://bioinformatics.psb.ugent.be/webtools/plantcare/html/, accessed on 21 February 2025) for cis-acting element prediction. Duplicate and erroneous entries were removed from the cis-element dataset, followed by the classification and quantification of the elements associated with each gene. A stacked bar chart illustrating the number of cis-acting elements was generated using GraphPad Prism 8.0 and subsequently merged with positional plots using Adobe Illustrator 2024. All combined figures presented in the article were finalized using Adobe Illustrator 2024.

### 4.5. Expression Patterns of CsTIFYs in Gender Differences and Adverse Stress

To preliminarily investigate the molecular functions of CsTIFY genes, two high-quality transcriptome datasets were downloaded from the NCBI GEO database (https://www.ncbi.nlm.nih.gov/geo/, accessed on 21 February 2025; PRJNA672722, PRJNA498707; Appendix A). PRJNA672722 represented the *C. sativa* transcriptomic data under alkaline stress. PRJNA498707 contains transcriptomic data derived from the glandular trichomes of female *C*. *sativa* flowers. Because of the comprehensive sample size and phenotypic data available in PRJNA498707, this dataset was selected for WGCNA. The remaining dataset was excluded from WGCNA because of its insufficient sample size. WGCNA was conducted using the WGCNA Shiny plugin in TBtools, following the protocol described by Wang et al. [85]. The expression data for *CsTIFY* genes were extracted from the aforementioned transcriptome datasets, and a gene expression heatmap was generated using the HeatMap function in TBtools.

### 4.6. Stress Treatment and RNA Extraction

In this study, the hemp variety we used was Longdama 6 (LDM6), which is a type of high-CBD medicinal hemp. Additionally, the THC content of LDM6 is significantly lower than 0.3%. The seeds were harvested from Kangjin town, Heilongjiang (126°48′ E, 46°11′ N), China. The seeds were sown in pots, with the following cultivation conditions: the temperature maintained at 25 °C, humidity maintained at 65%, and a light/dark cycle of 16 h of light followed by 8 h of darkness [86]. Eight-week-old plants of similar size were selected and divided into several groups. One group was subjected to alkaline treatment, where the plants were irrigated with a 200 mmol/L Na_2_CO_3_ solution (alkaline treatment group), while the other group received the same volume of water (control group). Apical leaf samples were collected 36 h after treatment and cryopreserved. Additionally, for the alkaline stress treatment, leaf samples were collected at 0, 12, 24, and 36 h and stored at −80 °C for further analysis. Total RNA was extracted from frozen leaf tissues using the Total RNA Extractor (Trizol) kit. Reverse transcription was performed using Maxima Reverse Transcriptase (Thermo Scientific, Shanghai, China) following the manufacturer’s instructions.

### 4.7. Primer Design and RT-qPCR Analysis

The CDS of CsTIFYs was obtained from the genome, and specific primers were designed using PRIMER PREMIER 6.0. The parameters were set as follows: a primer length of 18–24 bp, GC content of 40–60%, melting temperature of 55–60 °C, and amplicon length of 120–260 bp (Appendix A). In addition to the 14 TIFY genes, we designed four primers, each corresponding to a gene involved in cannabinoid metabolic pathways (*CsOLS*, *CsCBGAS*, *CsCBDAS*, and *CsTHCAS*). Based on the method described by Guo et al. [87], *CsEF1α* was used as the internal reference gene. RT-qPCR was performed using the 2× SGExcel FastSYBR Mixture (Sangon, Shanghai, China) on a Thermo Fisher Scientific 7500 Real-Time PCR System (ABI 7500 Software). Each 20 μL reaction consisted of the recommended reagent mixture, and the thermal cycling program was set as follows: initial denaturation at 95 °C for 3 min and 40 cycles of 95 °C for 15 s and 60 °C for 45 s, followed by a melt curve analysis. All reactions were performed in triplicate. Relative gene expression levels were calculated using the 2^−∆∆Ct^ method [88] and statistically analyzed using IBM SPSS Statistics version 26.0. The experimental design is illustrated in Appendix A.

## 5. Conclusions

TIFY transcription factors are known to play critical regulatory roles in secondary metabolism and stress responses. Although members of the TIFY family have been identified and functionally characterized in numerous crop species, limited information is available regarding TIFY transcription factors in *C. sativa*. In this study, 14 TIFY transcription factors were identified using bioinformatics approaches combined with transcriptomic data. These members were classified into four subgroups exhibiting distinct differences in gene structures, protein features, and predicted functions. Among them, the JAZ subfamily acted as a key suppressor of JA signaling. Consistent expression patterns between transcriptome datasets and RT-qPCR experiments confirmed the involvement of *CsJAZ5/6*, *CsPPD1*, and *CsTIFY1* in alkaline stress responses and cannabinoid metabolism. Moreover, *CsJAZ2/3/6* were implicated in the regulatory network of cannabinoids and warrant further investigation. However, the precise molecular mechanisms linking TIFY transcription factors to cannabinoid production remain largely unexplored. Future research should focus on elucidating the physiological functions of *CsJAZ5/6*, *CsPPD1*, and *CsTIFY1* in stress adaptation and cannabinoid accumulation. This study provides a foundational dataset supporting the potential involvement of TIFY transcription factors in the alkaline stress response and cannabinoid metabolism in *C. sativa*.

## Figures and Tables

**Figure 1 ijms-26-08171-f001:**
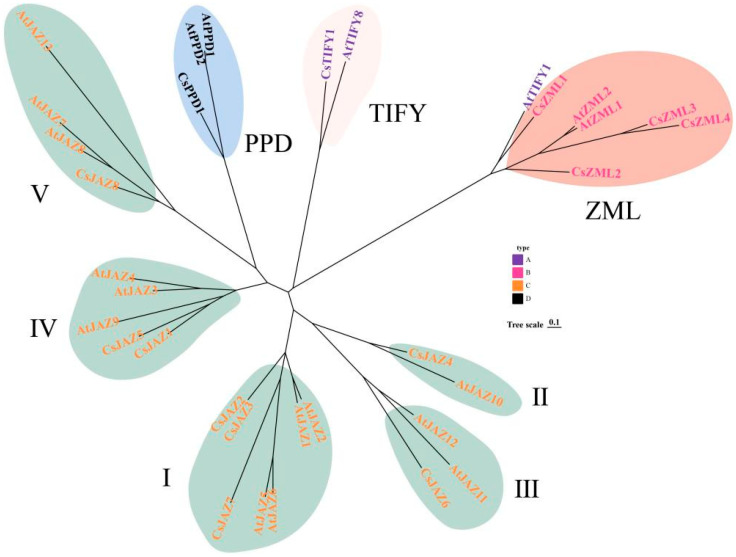
Phylogenetic tree of TIFY in *C. sativa* and *A. thaliana*.

**Figure 2 ijms-26-08171-f002:**
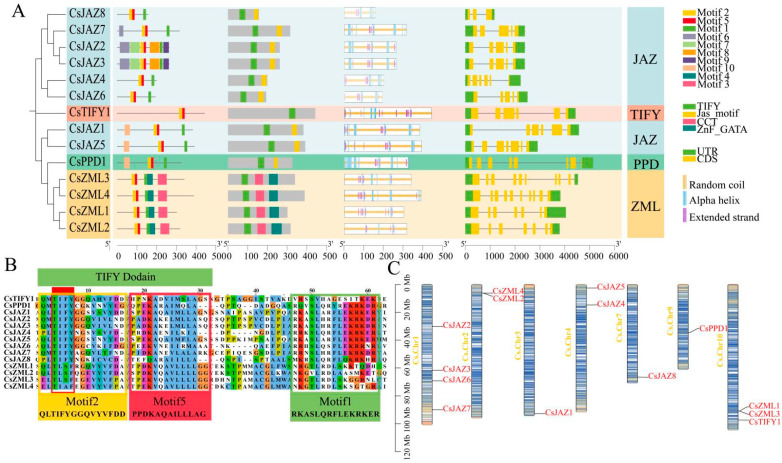
Gene structure and chromosomal localization of the TIFY family in *C. sativa*. (**A**) Gene structure and secondary protein structure. (**B**) Conserved motifs. (**C**) Chromosomal localization.

**Figure 3 ijms-26-08171-f003:**
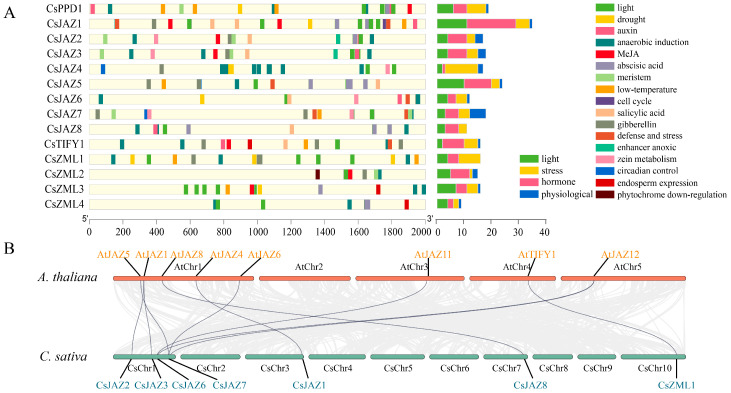
Cis-acting elements and collinearity analysis of TIFY in *C. sativa*. (**A**) Cis-acting elements. (**B**) Collinearity analysis.

**Figure 4 ijms-26-08171-f004:**
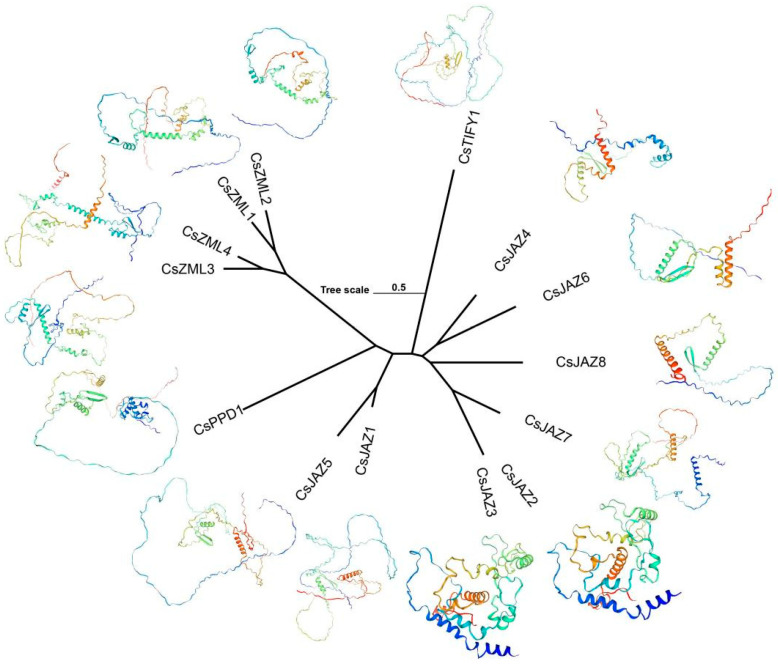
Protein structure of TIFY in *C. sativa*.

**Figure 5 ijms-26-08171-f005:**
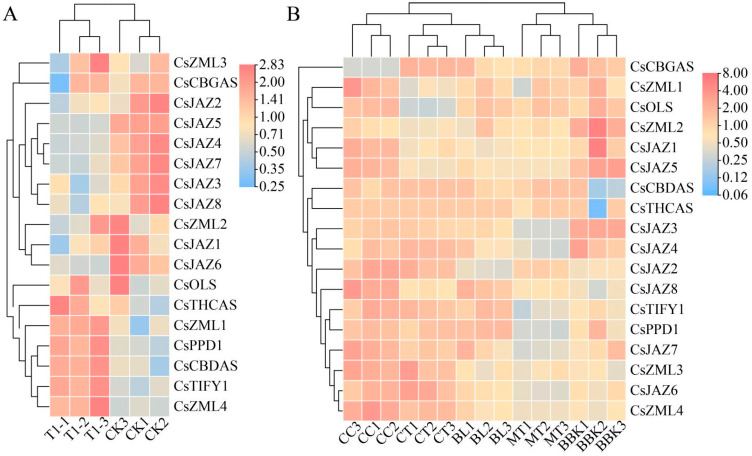
The expression pattern of *C. sativa* TIFY in the presence of alkali stress and in inflorescence tissue. (**A**) Differential expression of TIFY in *C. sativa* under alkali stress. T–CK1–3: control group (0 mmol/L NaHCO_3_); T–1–3: alkali treatment group (100 mmol/L). (**B**) Differential expression of TIFY in *C. sativa* inflorescence tissue. BBK1-3: Black Berry Kush; BL1-3: Black Lime; CT1-3: Canna Tsu; CC1-3: Cherry Chem; MT1-3: Mama Thai.

**Figure 6 ijms-26-08171-f006:**
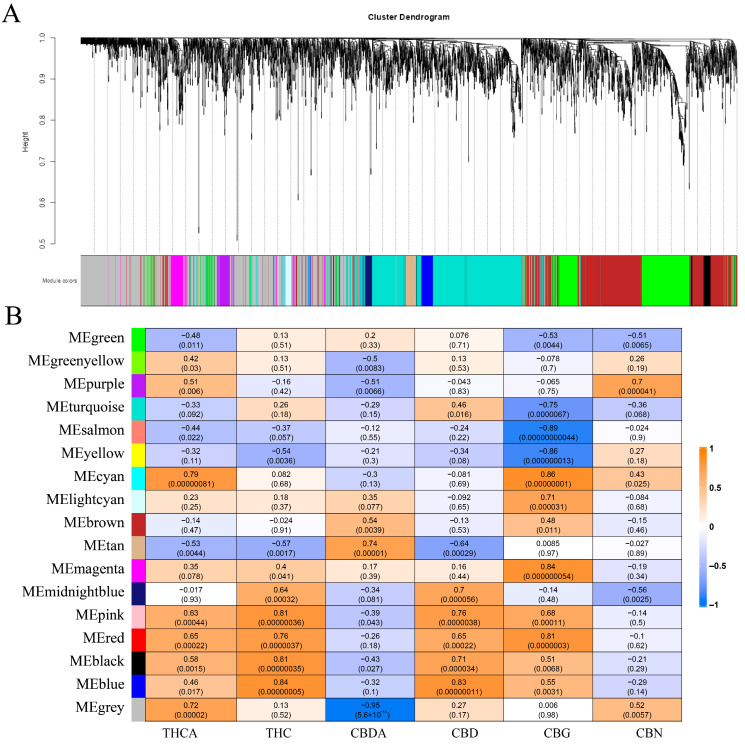
WGCNA of TIFY in *C. sativa*. (**A**) WGCNA modularity analysis of the *C*. *sativa* genome. (**B**) Association analysis between gene modules and cannabinoids.

**Figure 7 ijms-26-08171-f007:**
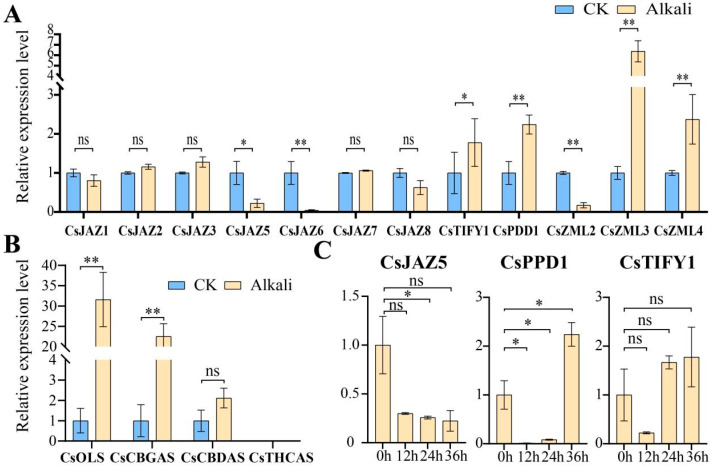
Relative expression levels of TIFY in C. sativa. under different treatments. (**A**) Relative expression level of CsTIFYs under alkaline stress. (**B**) Relative expression of 4 genes involved in cannabinoid metabolic pathways under alkaline stress. (**C**) Relative expression levels of CsJAZ5, CsPPD1, and CsTIFY1 under alkali stress at 0, 12, 24, and 36 h. * represents *p* < 0.05; ** represents *p* < 0.01; ns represents *p* > 0.05.

**Table 1 ijms-26-08171-t001:** Detailed information on 14 *CsTIFY* genes in *C. sativa* and their encoded proteins.

Name	Gene ID	Number of Amino Acid/Amino Acid	Molecular Weight/kDa	pI	Instability Index	GRAVY	Alpha Helix	Extended Strand	Random Coil
CsPPD1	LOC115723116	326	35,518.60	8.06	53.57	−0.728	21.78%	5.21%	73.01%
CsJAZ1	LOC115711241	382	40,204.84	9.35	53.22	−0.387	10.73%	4.71%	84.55%
CsJAZ2	LOC115706817	262	28,721.44	9.16	51.54	−0.528	14.89%	5.73%	79.39%
CsJAZ3	LOC115706226	262	28,672.37	9.18	50.96	−0.537	13.36%	6.11%	80.53%
CsJAZ4	LOC115712145	199	22,310.31	9.04	58.72	−0.459	14.57%	8.04%	77.39%
CsJAZ5	LOC115714189	391	42,034.53	8.74	49.53	−0.295	7.67%	5.63%	86.70%
CsJAZ6	LOC115705512	193	21,190.96	5.83	75.37	−0.577	9.84%	7.77%	82.38%
CsJAZ7	LOC115707122	315	34,251.27	8.93	49.35	−0.688	12.38%	4.13%	83.49%
CsJAZ8	LOC115696976	156	17,524.89	9.16	63.53	−0.552	17.31%	6.41%	76.28%
CsTIFY1	LOC115699142	442	46,479.54	9.13	51.16	−0.587	6.56%	7.01%	86.43%
CsZML1	LOC115697696	301	32,461.79	6.21	38.01	−0.654	4.32%	6.98%	88.70%
CsZML2	LOC115719706	317	33,897.48	5.57	34.85	−0.656	3.79%	3.15%	93.06%
CsZML3	LOC115697707	339	37,386.45	5.01	43.30	−0.703	7.67%	4.13%	88.20%
CsZML4	LOC115718519	388	42,163.76	4.75	49.51	−0.542	7.47%	4.12%	88.40%

## Data Availability

Data is contained within the article or Appendix A. The original contributions presented in this study are included in the article/Appendix A. Further inquiries can be directed to the corresponding author.

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
