# Peer review of "Genome-Wide Identification of the TIFY Family in Cannabis sativa L. and Its Potential Functional Analysis in Response to Alkaline Stress and in Cannabinoid Metabolism"

_ijms, 2025, doi:10.3390/ijms26178171_

Round 1

Reviewer 1 Report (New Reviewer)

Comments and Suggestions for Authors
  1. The title of the paper should be revised. It is not logical. " Expression analysis in alkaline stress and cannabinoid metabolism". Expression under alkaline stress and under cannabinoid metabolism is not a logical sentence structure.

Furthermore, it should be specified in the abstract that the study was conducted with young seedlings, and that the stress was a short duration stress.

  1. Line 51-56 includes numerous statements which are not backed up by references.
  2. A main problem of the manuscript is that the logical reasoning for studying effects of alkaline stress in cannabis is not well explained. Alkaline stress is not a main problem for cannabis research. It might be a local issue. If such it should be clearly defined as such.
  3. Another major issue is the developmental stage of the plants analyzed- young seedlings. It is well known that cannabinoids are mainly biosynthesized in cannabis in trichomes on female inflorescences. Therefore, the analysis of cannabinoid metabolism genes in young seedling may overlook effects on secondary metabolism in the appropriate developmental stage. This should be discussed and emphasized in the abstract' introduction, and the discussion.
  4. Another major issue is that the introduction focusses on the genes, but is severely lacking information about what is alkaline stress and how it affects plants. A paragraph should be added to the introduction about this.
  5. Furthermore, by now we know that numerous stress conditions have considerable effects on cannabis physiology and secondary metabolism. However, this issue is not mentioned al all in the manuscript. Nitrogen ( doi:10.1016/j.indcrop.2021.113516) and prosperous (doi:10.3389/fpls.2021.657323) deficiency stresses stimulate cannabinoid and terpene biosynthesis;  heat stress (doi:10.3389/fpls.2021.646020  ) stimulates cannabinoid production , salinity stress  (doi.org/10.17660/ActaHortic.2023.1377.97  ) reduces cannabinoid accumulation, and high ammonium/nitrate ratio stress reduced reduces cannabinoid and terpens production (oi:10.3389/fpls.2022.830224 ) . The introduction should include a short mention of these known stress effects.
  6. The M&M section should be improved to include a better description of the growing conditions, and the stress treatment.

Author Response

Reviewer 2 Report (New Reviewer)

Comments and Suggestions for Authors

The article provided comprehensive analysis of CsTIFY gene family and useful information were also provided in the gene function involved in alkaline stress response and cannabinoid metabolism. What I concerned is that the genome-wide identification and characterization of TIFY gene family in Cannabis sativa had been reported by Wen et al. (2020) in Chinese Journal of Experimental Traditional Medical Formulae. Based on the published references, there were also 14 members were identified and the physicochemical properties, phylogenetic trees, gene structures, chromosome locations and gene expression patterns were analyzed. The major work in the present study is just the same as the reference. So, how did the authors think about the novelty and scientific soundness of the present study? Besides, some suggestions are listed below.

  1. Line 4: Is “Zhang Ming and Fang Yuyan” one person? There should be spaces behind the commas.
  2. Line 37: The number of references should follow the Journal’s rules.
  3. Line 44: the Latin name should be given when the species showed at the first time.
  4. Line 47: Correct format of letters should be used when gene or protein name were mentioned.
  5. Line 65: Abundant functions have been mentioned in the paragraph, however, there were little information about the roles in alkaline stress. The information is important for setting the hypothesis of the present study.
  6. Line 103: Do the authors mean margin land instead of non-staple crop land?
  7. Line 108:
  8. Section 2.1: the Pan-genome of Cannabis has been released this year. Did the author use the nearest data base for this study?
  9. Section 2.7: Cannabinoid biosynthesis primarily occurs in the glandular trichomes of the flower buds. Why the RT-qPCR analysis used only leaf tissue without data from flower buds?
  10. Section 3.1: The basic characteristics of CsTIFYs were different from the reference Wen et al., 2020, including the length of coding sequences, the chromosomal localization and others. How to explain the difference?
  11. The GMQE values for the protein tertiary structure modeling (0.45–64) are generally below the credible threshold (>0.7), and the sequence identity for some models is only 51–52% (e.g., CsJAZ2/3), which affects the inference of structure–function relationships (Section 3.5).
  12. Figure 4. The figure is not enough clear to read.
  13. Figure 6. WGCNA showed that CsJAZ2/3/6 were correlated with CBD/THC content, but correlation does not imply causation. Is this an over-interpretation?
  14. Figure 7. The figure showed that CsPPD1, CsTIFY1 were significantly inhibited in 12 h under alkali stress but triggered thereafter. The authors should give a explanation for such phenomenon not only description.
  15. Line 542. The Latin name of species should be italic.
  16. The format of references does not match the requires of the Journal.
Comments on the Quality of English Language

The quality of English could be improved.

Round 2

Reviewer 1 Report (New Reviewer)

Comments and Suggestions for Authors

The manuscript was improved. However, there are issues with the references in the manuscript which must be corrected. Specifically, I noted that in the new paragraph added in lines 134-145 the numbers of the citations in the text do not represent the appropriate manuscripts in the reference list. For example, [47] should be [48] and so on. Furthermore, in the reference list one manuscript appears twice, no. 49 and 51 are the same manuscript. I believe that one of this manuscript was meant to be  doi.org/10.1016/j.indcrop.2024.119157  which demonstrates effect of flushing (osmotic stress).

Author Response

Reviewer 2 Report (New Reviewer)

Comments and Suggestions for Authors

The author has addressed the questions raised about the the present study, provided sufficient evidence, and supplemented materials. It is recommended to accept the paper.

Comments on the Quality of English Language

There are still some format mistakes.

Author Response

This manuscript is a resubmission of an earlier submission. The following is a list of the peer review reports and author responses from that submission.

Round 1

Reviewer 1 Report

Comments and Suggestions for Authors

The paper describes genome-wide identification and expression pattern analysis of the TIFY gene family in hemp. There are lots of questions/issues that must be resolved, so the paper can not meet the standard to be published in International Journal of Molecular Sciences.

  1. In the Introduction, I do not know why the authors identify the TIFY gene family in hemp, especially why use 5 sets of hemp transcriptome data in the GEO Datasets. The research purpose is unclear.
  2. In the Materials and Methods, where were the genome of Arabidopsis and hemp download from NCBI, what is the website address? And genome number?
  3. Five transcriptome data (PRJNA756306, PRJNA672722, PRJNA1108719, PRJNA1199007, PRJNA498707)should be explained in the Materials and Method. Each data should introduced the background and basic information.
  4. 14 TIFY geneswere identified in hemp genome, but how to name them? Why their names are listed like in Table 1. Please give some reasons.
  5. In the table 1, what AA/aa means? And MW? II? Their meanings are not clear. Please give some notes.
  6. Many figures are too small, I can not see them clearly. Such as Figure 2 and Figure 5 D.
  7. In 6. Stress expression heatmap of TIFY transcription factors, but the transcriptome data of male and female differences in wild hemp was not stress. And where is the figure or table of hemp inflorescence transcriptome data?
  8. In the Discussion, there is no reference in the 4.1. There is only some analysis of the results, but no discussion. So the discussion is insufficient.
  9. In the Conclusions, the authors should give some important results, especially some results of expression analysis and some candidate gene.
  10. There should be some gene expression analysis of stress treatment by RT-qPCR.
  11. There are a lot of grammar errors and mistakes in the whole manuscript. For example, Latin names and gene names need to be italicized; A space is required after the full stop. Some sentence expressions are unclear, such as, AtTIFY is a TIFY in Arabidopsis in Figure 1; and A: Male and female differences in wild hemp. B: Effects of alkali stress. C: Effect of Trichoderma hamatum on drought resistance. D: different stages of hemp infection with Sclerotinia sclerotiorum in Figure 5. These sentences are incomplete.
  12. Hemp should be changed into Cannabis sativa L. in the whole text. Because there are some wild materials.
  13. The reference format is not standardized.

Reviewer 2 Report

Comments and Suggestions for Authors

Comments on the Quality of English Language

English language needs improvement

Round 2

Reviewer 1 Report

Comments and Suggestions for Authors

Thank you for your revised version, but there are some questions need to be revised before the manuscript can be published.

  1. I still do not know why the authors identify the TIFY gene family in hemp? The research purpose is unclear. Because there many purpose in this manuscript, such as hempunder adverse stress conditions, cannabinoids in hemp, and hemp treated by JA, and TIFYs in different gender plant. It will make the readers very confused. There is no unified theme. Please reorganize the content.
  2. The Latin names of species (not only hemp) in this manuscript are also not correct. Latin names should be italicized. Take Cannabis sativa L. for example, Cannabis sativa should be italicized, but not L.. First occurrence is Cannabis sativa L., but after that, it should be C. sativa. A space is required after the full stop.
  3. Why only choose the inflorescence transcriptome data to make a WGCNA analysis? But in the RT-qPCR, there is no gene expression verification for it.
  4. In Figure 5, there is no notes in the Fig. For example, what is T1, T2, and T3? What is TDRT1\2\3? What is 20hUTC? .... And the title ‘Stress expression heatmap of TIFY in Cannabis sativa L.’is not correct, ‘Male and female differences in wild hemp’ is not stress expression. ‘B:Effects of alkali stress’ is not complete.
  5. In Figure 7, A and B are not correct. The colour of CK and Alkali, CK and MeJA are the same, it can not be distinguished. There is no analysis of variance between treatments.
  6. The discussion is insufficient, there is only very few reference in each part

Reviewer 2 Report

Comments and Suggestions for Authors

Dear author,

After thoroughly checking, I found it is improved significantly.

Round 3

Reviewer 1 Report

Comments and Suggestions for Authors

Thank you for your revised version, but there is still a very important question that the authors can not resolve.

Why the authors identify the TIFY gene family in hemp? The purpose in this paper is unclear. Because there are many experiments in this manuscript, such as hemp under adverse stress conditions, cannabinoids in hemp, and hemp treated by JA, and TIFY genes in different gender plant. It will make the readers very confused. So, please determine the purpose of the research firstly, and then resubmit the manuscript.